# 3D Reconstruction with Coronary Artery Based on Curve Descriptor and Projection Geometry-Constrained Vasculature Matching

Jijun Tong, Shuai Xu, Fangliang Wang and Pengjia Qi *

School of Information, Zhejiang Sci-Tech University, Hangzhou 314423, China; jijuntong@zstu.edu.cn (J.T.);
201920602011@mails.zstu.edu.cn (S.X.); 201820602012@mails.zstu.edu.cn (F.W.)
* Correspondence: qipengjia@zstu.edu.cn

**Abstract:** This paper presents a novel method based on a curve descriptor and projection geometry constrained for vessel matching. First, an LM (Leveberg–Marquardt) algorithm is proposed to optimize the matrix of geometric transformation. Combining with parameter adjusting and the trust region method, the error between 3D reconstructed vessel projection and the actual vessel can be minimized. Then, CBOCD (curvature and brightness order curve descriptor) is proposed to indicate the degree of the self-occlusion of blood vessels during angiography. Next, the error matrix constructed from the error of epipolar matching is used in point pairs matching of the vascular through dynamic programming. Finally, the recorded radius of vessels helps to construct ellipse cross-sections and samples on it to get a point set around the centerline and the point set is converted to mesh for reconstructing the surface of vessels. The validity and applicability of the proposed methods have been verified through experiments that result in the significant improvement of 3D reconstruction accuracy in terms of average back-projection errors. Simultaneously, due to precise point-pair matching, the smoothness of the reconstructed 3D coronary artery is guaranteed.

**Keywords:** 3D reconstruction; parameter-adjusting LM algorithm; curve descriptor; dynamic programming; surface reconstruction

## 1. Introduction

A large number of humans die from coronary artery disease every year. An effective way to check for coronary artery disease is coronary angiography. XRA (X-ray angiography) has better imaging quality and a faster imaging speed than alternative imaging methods such as CTA (computed tomography angiography) and magnetic resonance angiography. However, the 3D spatial structure of the blood vessels superimposed on a 2D image may overlap each other in the contrast image and obstruct the doctor's observation, which is the main disadvantage of X-ray imaging. Reconstruction of the contrast image can fully mine the information of the contrast image and obtain the best perspective from the 3D reconstructed image, which can effectively improve the accuracy of the quantitative analysis of clinical medical parameters [1,2], estimate the vascular motion parameters, and study the condition of the heart movement. This technique has great clinical value.

The 3D reconstruction of the coronary arteries is achieved by using two planar angiographic images of the same blood vessel segment from different angles. The beating of the heart, the error of the two-dimensional image extraction, and the error of the system parameters affect the accuracy of the 3D reconstruction. By optimizing the spatial relationship between the planar angiography, photos can improve the reconstruction accuracy, and the spatial relationship is characterized by rotation and translation transformation.

It is mandatory to accurately find the corresponding points of the blood vessel between the two images and then optimize the system parameters to achieve a 3D reconstruction. A challenging problem is to find the corresponding points in another image.

Therefore, our method uses two uncalibrated 2D cardiovascular images to achieve the 3D reconstruction of blood vessels. First, we extract the centerline of the blood vessel and measure its radius. Then, utilizing the information stored in the image file header along with a set of corresponding characteristic parameters, we optimize the internal and external parameters by employing the parameter-adjusting LM algorithm. The projection matrix of each view will be constructed by parameters refinement. Since the corresponding relationship of each point of the blood vessel between the two views is not obvious, therefore, under epipolar constraints, a cost matrix of potential matching points is constructed. Before applying dynamic programming to solve the optimal matching path of the cost matrix, it is required to calculate the CBOCD descriptor of the centerline to indicate the degree of distortion and overlap of the centerline. It can be regarded as the step size of different stages of dynamic programming. After the matching point pairs are obtained, the 3D reconstruction of the coronary arteries can be performed according to the binocular stereo vision theory in computer vision, and a set of unorganized points generated by the normal radius can be used to complete the implicit surface reconstruction. Here are our contributions to this paper:

- We proposed a descriptor called CBOCD which is constructed curved shape with pixel brightness. It can characterize the distortion and overlap of the current point on the curve during projection.
- We used epipolar line constraint to construct the matching cost matrix, and applied the CBOCD descriptor as the step size in the dynamic programming method to find the best matching path globally.
- We used the PALM algorithm combined with the trust region method to optimize the matrix of geometric transformation.

## 2. Related Work

In the past few years, many algorithms for 3D modeling methods based on two 2D images have been proposed. These algorithms are generally divided into two categories: 3D modeling from calibrated data [3,4] and 3D modeling from uncalibrated data [5,6]. A single calibration step, which can determine the geometry of the image and eliminate image distortion, is the main difference between the two categories. In this article, we discuss uncalibrated data.

### 2.1. Geometric Transformation Optimization

The error of the system parameters can be reduced by the system calibration method, but this method further complicates the process and requires additional template calibration [7]. The adaptive simulated annealing method has a fast convergence speed and good robustness. It is a heuristic algorithm, but its results are random and cannot be used in medical diagnosis that requires stability and accuracy [8]. Currently, the most used algorithm is the Levenberg–Marquardt algorithm, which can be used for parameter optimization [9], but it needs to improve the speed of convergence and global convergence.

### 2.2. Point-Pairs Matching

The traditional method is to use epipolar constraint to search matching point pairs of blood vessels in two images [10–14]. In these methods, there are inherent limitations and errors on the epipolar constraint, which will lead to point-pair mismatches. Therefore, the point-pair mismatches are limited by some constraints.

### 2.3. Curve Descriptor

To ensure the smoothness of the curve and accuracy of matching at the same time, more features and constraints should be adopted. Freeman [15] used the starting coordinates of the curve and the direction code values of adjacent pixels to express the curve. This method uses chain code to represent the curve. Wang [16] proposed the MSLD (mean–standard deviation line descriptor) based on the idea of the neighborhood location division of SIFT.

The invariant features of each sub-region of all pixels on the curve are counted. Then he extended it to curve matching and got the MSCD (mean–standard deviation curve descriptor). The descriptor successfully solves the problem of the uniform description of different length curves. It is the most representative curve texture descriptor in recent years. Shape context is proposed to describe the geometric characteristics of a curve, which map the relationship between sampling points and relative positions by optimizing statistical histograms [17].

In this paper, the parameter-adjusting LM algorithm combined with the trust region method is adopted, and the cost matrix of the possible matching point pairs is calculated by constructing the epipolar constraint to greatly reduce the number of iteration steps and converge the error to an ideal value.

## 3. Methodology

### 3.1. Epipolar Line

As shown in Figure 1, calculating a 3D point coordinate from the corresponding points on two 2D images is a dual-view geometry problem, which can be constrained by geometric relations.

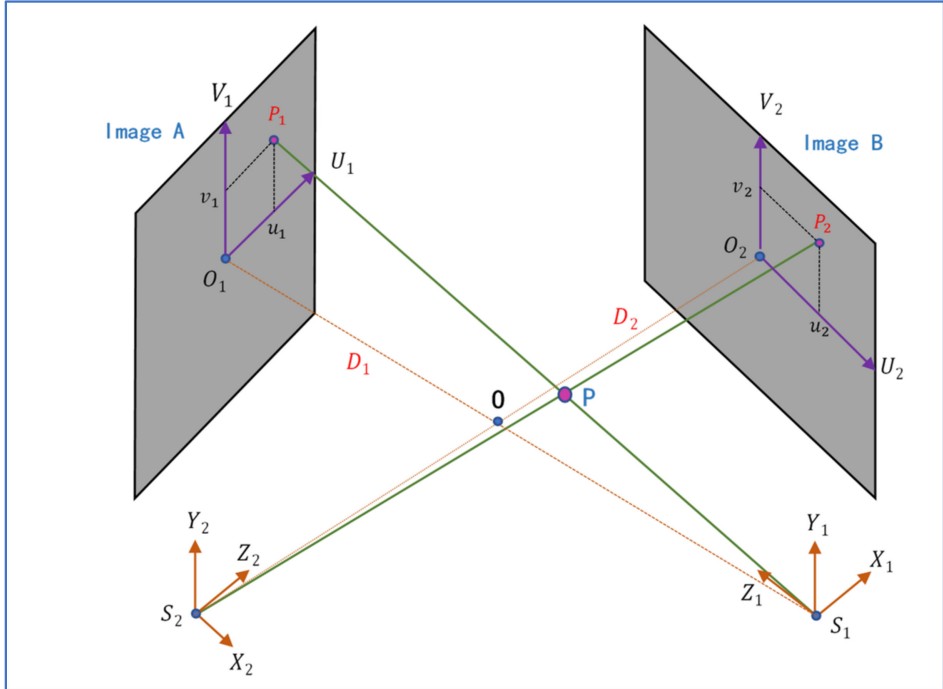

**Figure 1.** The geometric relationship between two coronary angiography images.

The 3D coordinates with the geometric transformation relationship defined in the coordinate system $X_1Y_1Z_1S_1$ and the 3D coordinates $(x_2, y_2, z_2)$ defined in the coordinate system $X_2Y_2Z_2S_2$ can be expressed by

$$[x_2, y_2, z_2]^T = R \cdot \left( [x_1, y_1, z_1]^T - \vec{t} \right),$$
$$\left( R \text{ is the rotation matrix; } \vec{t} \text{ is the translation vector} \right). \tag{1}$$

For a 3D point P, the following formula can solve its coordinates $(x_1, y_1, z_1)$ in the $X_1Y_1Z_1S_1$ coordinate system:

$$
\begin{bmatrix}
1 & 0 & -\xi_1 \\
0 & 1 & -\eta_1 \\
a_1 & a_2 & a_3 \\
b_1 & b_2 & b_3
\end{bmatrix}
\cdot
\begin{bmatrix}
x_1 \\
y_1 \\
z_1
\end{bmatrix}
=
\begin{bmatrix}
0 \\
0 \\
\vec{a} \cdot \vec{t} \\
\vec{b} \cdot \vec{t}
\end{bmatrix},
\tag{2}
$$

$$
\vec{a} = [R_{11} - R_{31} \cdot \xi_2 \;\; R_{12} - R_{32} \cdot \xi_2 \;\; R_{13} - R_{33} \cdot \xi_2];
$$
$$
\vec{b} = [R_{21} - R_{31} \cdot \eta_2 \;\; R_{22} - R_{32} \cdot \eta_2 \;\; R_{23} - R_{33} \cdot \eta_2]
$$

The components of the rotation matrix are represented by $R_{ij}$ $(i, j = 1, 2, 3)$. The equation set consisting of four linear equations, which can be abbreviated as $A \cdot C = B$, used to solve the $x_1$, $y_1, z_1$, is a set of over-limit equations. The solution of the least-squares solution is as follows:

$$
C = \left(A^T \cdot A\right)^{-1} \cdot A^T \cdot B,
\tag{3}
$$

As shown in Figure 2, the epipolar constraint limits the relationship of a 3D point on two projection planes.

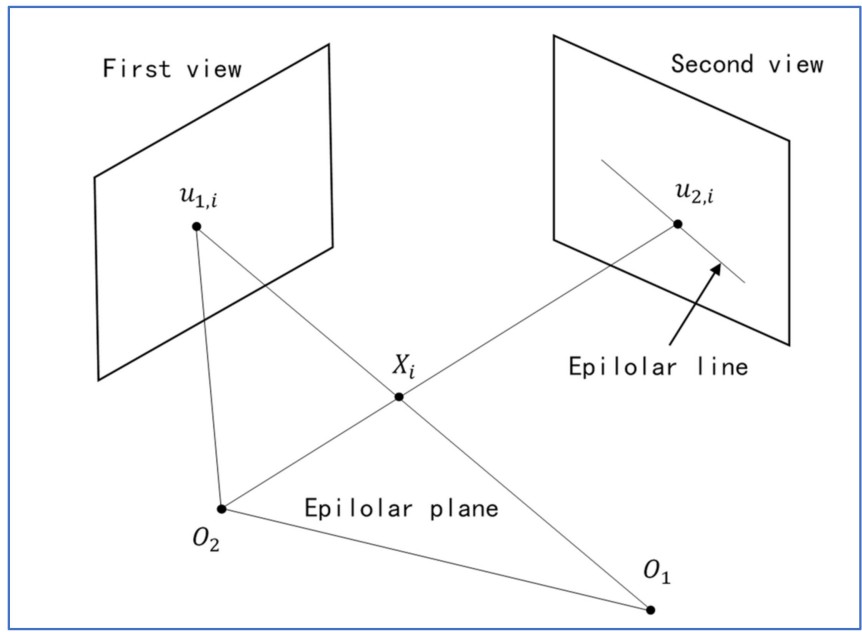

**Figure 2.** Epipolar constraints for two angiograms.

The calculation process of the epipolar line is as follows:

$$
\begin{bmatrix} \xi_2 & \eta_2 & 1 \end{bmatrix} \cdot
\begin{bmatrix}
a_3 \cdot b_2 - a_2 \cdot b_3 \\
a_1 \cdot b_3 - a_3 \cdot b_1 \\
a_2 \cdot b_1 - a_1 \cdot b_2
\end{bmatrix}
= 0,
\tag{4}
$$

This is the epipolar line of $(u_1, v_1)$ in the First view, while $(u_2, v_2)$ in the Second view should satisfy this formula.

### 3.2. Geometric Transformation Optimization

This paper studies the reconstruction of specific vascular segments. The doctor manually marked the narrow blood vessel segment of interest. The two ends points of the selected blood vessel are usually the endpoints or branch points of the blood vessel [18,19]. These points are treated as a set of corresponding features to minimize the distance between

the centerline point and the reconstructed 3D centerline point of the corresponding back projection by constructing an objective function. The objective function is as follows:

$$E_i(GT) = \left( \|P'_{li} - P_{li}{}^2\| + \|P'_{ri} - P_{ri}{}^2\| \right), \tag{5}$$

Between the imaging coordinate systems, the geometric positional relationship during the two comparisons is recorded by the geometric transformation matrix described by $GT = \begin{bmatrix} R & \vec{t} \end{bmatrix}$. The i represents different corresponding feature points. $P_{li}$ is the point of the selected feature on the left image, and $P_{ri}$ is the corresponding point of $P_{li}$ on the right contrast image. The corresponding back projections of the reconstructed 3D centerline points are $P'_{li}$ and $P'_{ri}$. $P\| \|^2$ represents the second-order norm, and $\|P'_s - P_s\|^2$ represents the Euclidean distance between $P'_s$ and $P_s$.

The objective function $E$ described in Equation (5) is a nonlinear least-squares problem. The PALM (parameter adjustment LM algorithm) used in this paper was proposed by Qi Yanli [20], and its convergence speed and robustness are within an ideal range.

The LM parameter is dynamically adjusted by the ratio of the actual decline to the predicted decline. Convergence and convergence speed is improved by this method.

*3.3. CBOCD*

Our goal was to find out the vascular segments that are overlapping or distorted during projection in angiographic images. It is called OPE (overlapping or parallel elements) in this paper. As shown in Figure 3, several features of OPE can be summarized:

- The brightness of the pixel where OPE is located is often lower than the adjacent anterior and posterior pixels.
- OPE is often on the convex arc of the blood vessel.

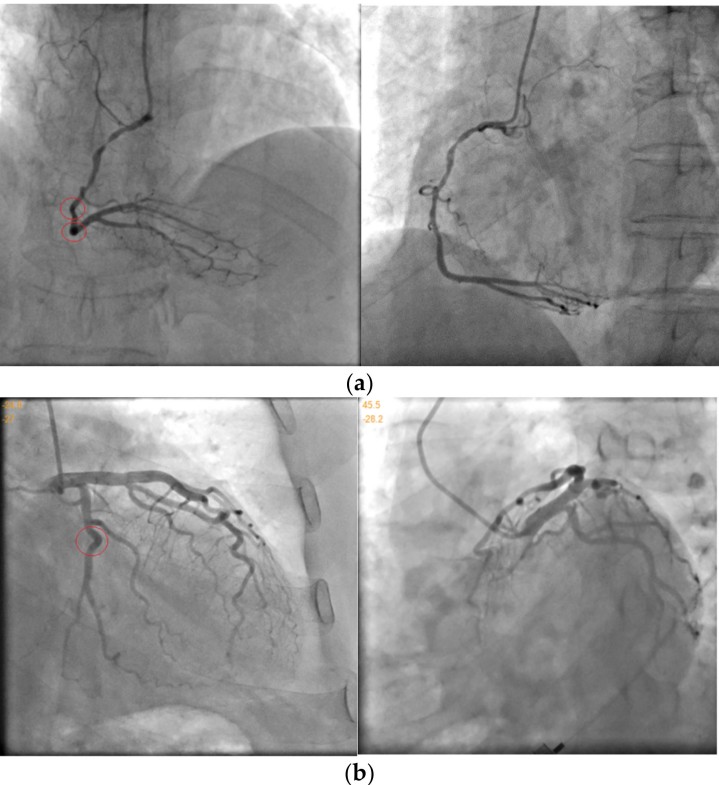

**Figure 3.** The example of OPE. (**a**,**b**) are two sets of patient data. The red circle on the left image is where OPE is; the image on the right is the corresponding.

We chose to connect adjacent convex arcs with short distances on blood vessels as OPE candidate sets named Curvature OPE constructed from curvature. Then, the CBOCD descriptor is proposed based on the possible contribution value of brightness and curvature to OPE. The specific definition is as follows:

$$
\begin{aligned}
&\text{CBOCD}_i = 5, \ P_i \in \ Brightness \ ODE \ and \ Curvature \ ODE \\
&\text{CBOCD}_i = 4, \ P_i \in \ Curvature \ ODE \\
&\text{CBOCD}_i = 3, \ P_i \in \ Brightness \ ODE \\
&\text{CBOCD}_i = 2, \ other
\end{aligned}
\tag{6}
$$

The specific steps of constructing the CBOCD descriptor of the curve are as follows.

### 3.3.1. Curve Preprocessing

Usually, the blood vessels of the standard image are longer than the blood vessels of the constructed image.

It is necessary to use NURBS (non-uniform rational basis splines) [21] for interpolation to increase the number of points on the blood vessels in the constructed image and reduce the difference with the standard image. Additionally, the longer vessel also needs to be fitting by NURBS to reduce the error of extracting the centerline and achieve sub-pixel accuracy.

### 3.3.2. Construction of Brightness OPE

To reduce the true content of the contrast agent contained in blood vessels, we assumed that the contrast medium attenuates or enhances uniformly from the source to the end of the vessel.

We calculated the average brightness of the first ten points as the brightness of the blood vessel source named InFirst, and calculated the average brightness of the last ten points as the brightness of the blood vessel source named InLast.

$$
InFirst = \sum_{i=0}^{9} Pixel_i \ InLast = \sum_{i=0}^{9} Pixel_{NUM-i}
$$

$Pixel_i$ is the brightness of the ith pixel on the centerline of the blood vessel. *NUM* is the points number on blood vessels in the Standard Image.

Because the curve is obtained by increasing the parameters on the spline during fitting, the distance between its adjacent points is almost the same. Therefore, we directly chose to use the following formula to determine whether the point belongs to Brightness OPE.

$$
InNow_i = InFirst + i \times \frac{InLast - InFirst}{NUM - 1}, i = 0, 1 \ldots NUM - 1,
\tag{7}
$$

If the brightness of the ith pixel is less than $InNow_i$, the pixel belongs to Brightness OPE.

### 3.3.3. The Construction of Curvature OPE

The way of constructing Curvature OPE is to identify all convex arcs on the curve. We need to connect the two adjacent convex arcs.

This paper proposes a convex arc recognition method suitable for this application scenario referring to the shape context and Freeman chain code.

To enhance the robustness to noise points, the algorithm, which is different from the methods only considering the curvature of a point, defines a pixel support region (PSR) for a pixel $P_i$ on curve L. As shown in Figure 4a, it defines a circle region centered at $P_i$ as PSR. Then the PSR is divided into $\alpha$ sectors of equal radian denoted as a block. We marked the blocks as $0, 1 \ldots \alpha - 1$ anticlockwise. We selected $\beta$ points before and after the current point, respectively, and then calculated the indication of the block to which the points belong. So,

every point has a $2\beta$ vector $\textbf{\textit{B}}$, denoted as $\textbf{\textit{B}}_i$. Then the union of all these PSRs is called the curve support region (CSR). The full set of vectors is also a rich description.

Then, we identifed whether $P_i$ belongs to Curvature OPE by analyzing $\textbf{\textit{B}}_i$. We needed to merge the blocks whose indications are diagonally opposite to each other into the same block and record them as smaller indications. Then, the number of indications of blocks was reduced to half of the original. We counted the number of times each indication of blocks appears in $\textbf{\textit{B}}_i$. If an indication appears with a frequency of $2\beta - \gamma$, it can be regarded as a straight line. If no indication meets the adjustment requirements, it is regarded as a point of Curvature OPE. In the first and last part of the curve, $2\beta$ was replaced by the number of points that it counted. $\gamma$ is used to control the degree of tolerance to noise. The larger $\gamma$ is, the stronger the tolerance to noise is.

As shown in Figure 4b. This algorithm is easy to produce ambiguity if the curve is close to the boundary of the block. Therefore, the ambiguity can be eliminated by rotating the current PSR clockwise by $180/\alpha$ angle. Then, we calculated $\textbf{\textit{B}}_i$ and analyzed it again to compare with the original results. Only the point that satisfies both two analyses can be regarded as one of the Curvature ODEs.

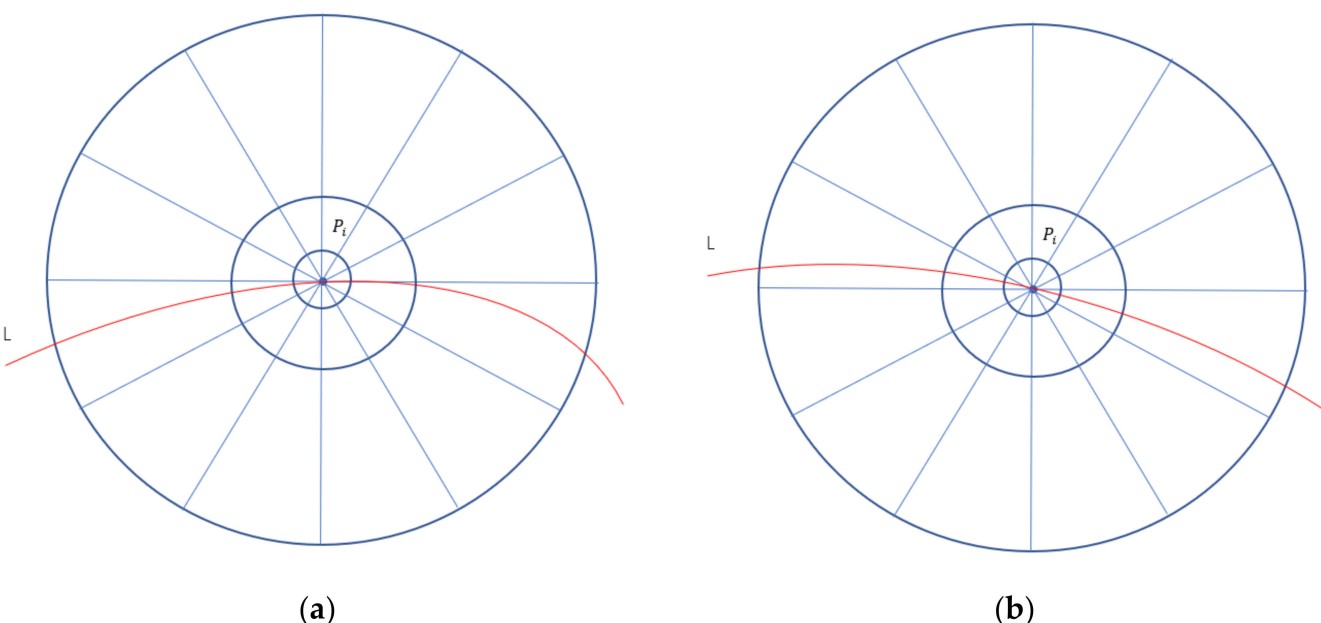

(**a**)  (**b**)

**Figure 4.** (**a**) is CSR of CBOCD, (**b**) is Ambiguity of (**a**).

The next step is to post-process Brightness OPE and Curvature OPE. Firstly, we connect the two parts of OPE that are relatively short apart, then delete the parts of OPE that are too short, and finally connect the two parts of OPE that are relatively long apart.

Finally, the CBOCD value of each point is determined by the Formula (6). So far, CBOCD descriptor of the whole curve has been constructed.

### 3.4. Dynamic Programming

This paper constructs a matrix of matching cost for all possible matching point pairs of the blood vessel and obtains the best matching point pair by solving the optimal solution of the cost matrix through dynamic programming. Special attention should be paid to the smoothness of blood vessels and the continuity of acupuncture points. The proposed method is as follows.

#### 3.4.1. Construct Vascular Matching Cost Matrix

On the standard image with N points on the blood vessel, "COST" is a matching cost matrix of N × N. $n_{ij}$ is the matching cost of the ith point on standard blood vessel image

and the jth point on the constructed blood vessel image, which is defined as the distance between the epipolar line generated by the ith point in the standard blood vessel image and the jth point in the constructed blood vessel image.

$$n_{ij} = \frac{|ma + nb + c|}{\sqrt{a^2 + b^2}}, \tag{8}$$

The Formula (8) is composed of the epipolar line described by $ax + by + c = 0$ on the standard image and the jth corresponding point on the structure image with coordinates $(m, n)$.

### 3.4.2. Optimal Path

The error of the current system parameters causes some points on the standard image to deviate from the epipolar line so that the epipolar line does not intersect the blood vessel. More precise system parameters can be obtained by finding the point pair with the largest distance between the epipolar line and the blood vessel.

The searching method is as follows.

First, the minimum value in each row of COST described as $MIN_i$ ($i = 0, 1 \ldots N, N$ is the number of vascular matching point pairs) are recorded.

Then, find the maximum value $MAX_{ij}$ of All $MIN$.

The highest cost matching point pairs in the standard image from all other ideal matching point pairs is denoted as $(P_i, Q_j)$, where $P_i$ is the point in the standard image and $Q_j$ is the point in the constructed image.

Finally, if $MAX_{ij}$ is bigger than the setting separation standard threshold STD, it can be said that the epipolar line does not intersect the blood vessels, and the point pairs $(P_i, Q_j)$ of the blood vessels are regarded as the deviation point pairs.

To minimize the global error caused by the optimization transformation when matching conjugate point pairs, the deviation point pairs will be added to the corresponding feature to optimize the transformation again to correct the defects of the old optimization transformation applied to the current feature.

Calculate the global shortest path from the upper left corner to the lower right corner of the matrix in COST to obtain the global minimum matching cost from the beginning to the end of the blood vessel.

The cost of different points in the matrix is a state variable in dynamic programming used to solve the shortest path. The point in the lower right corner of the matrix is used as a sub-stage decision-making problem to be solved, which is a boundary condition. The key to the solution is to determine the recursive relationship between this state and its next state. Therefore, the forward and backward stages of dynamic programming are set under the constraints of the blood vessel sequence. The settings are as follows:

First, because the shorter vessel is increased by interpolation using NURBS on the Constructed Image, the smoothness of the vessel on the Constructed Image is inferior to the smoothness of the vessel on the Standard Image. Therefore, it is necessary to sample on the standard image, and then find the point corresponding to the sampling point on the structure image. $N_{i+1}$ is the next line of $N_i$.

Second, the CBOCD descriptor built above will be used as the step size of each move in dynamic programming. The previous dynamic programming has a fixed step size for each state transition, which does not conform to the different degrees of scaling and elongation of angiogenesis at different angles during projective transformation. In this paper, CBOCD descriptor is used to describe the possibility that the current point belongs to OPE. The greater the possibility is, the larger the step size of state transition of dynamic planning at the current point is. Therefore, the adaptive step size of dynamic programming is realized in this paper.

If there is conjugate point pairs [i,j], it means that the ith point of the vessel on the Standard Image is corresponding to the jth point of the vessel on the Constructed Image.

The next conjugate point pairs have some possibility decided by the CBOCD descriptor of the Constructed Image and the Standard Image:

$$[i+1, j+m], m \in (0, \text{CBOCD}_i), \tag{9}$$

There is an exceptional case that if the number of j in calculated conjugate point pairs is equal to $\text{CBOCD}_j - 1$, the $m$ cannot be zero. $\text{CBOCD}_i$ belong to the Standard Image, and $\text{CBOCD}_j$ belong to the Constructed Image.

It is illustrated with a 4 * 5 template in COST as Figure 5.

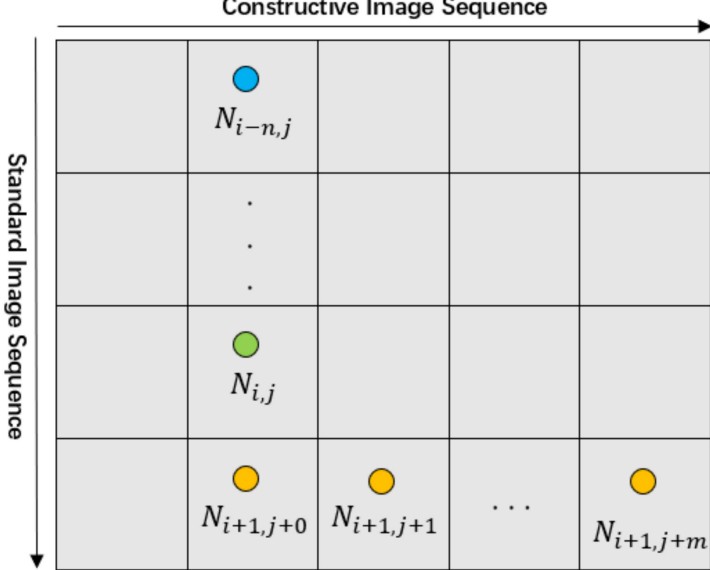

**Figure 5.** Diagram of forwarding and backward state recursion in dynamic planning.

The basic idea of dynamic programming is to calculate the best path from the initial position to all remaining positions in COST. The specific process is as follows:

First, construct a matrix "TOTAL" of cumulative optimal cost initialized with MAX. and a record precursor matrix "MATCH" initialized with $-1$. The initial values all mean that the elements cannot be reached. Assign the first line of TOTAL at start, and let $TOTAL_{0,j} = COST_{0,j}, (j = 0, 1, \ldots, N-1)$; assign the first column of TOTAL, and let $TOTAL_{i,0} = TOTAL_{i-1,0} + COST_{i,0}, (i = 1, 2, \ldots, CBOCD_0 - 1)$. The $CBOCD_0$ belong to the Constructed Image.

Second, there is a recursive formula from the second line as follows:

$$TOTAL_{i,j} = \min(TOTAL_{i-1,j}, TOTAL_{i-1,j-1} \ldots TOTAL_{i-1,j-m}) + COST_{i,j},$$
$$(i = 1, 2, \ldots, N-1, \ j = 1, 2, \ldots, N-1, m = CBOCD_{i-1}) \tag{10}$$

In particular, it needs to be noted that if the number of *j* in calculated conjugate point pairs is equal to $CBOCD_j - 1$, the $TOTAL_{i-1,j}$ cannot be chosen by regrading it as MAX. $CBOCD_{i-1}$ belong to the Standard Image, and $CBOCD_j$ belong to the Constructed Image. At the same time, the shortest path exists between the starting position and position $(i, j)$, and the precursor of the position $(i, j)$ on the shortest path is $MATCH_{i,j}$.

Third, obtain the cumulative optimal cost of the end element and the optimal path from the end element to the initial element through the recursive relationship.

Third, obtain the sum of the optimal cost with the endpoint and the optimal path from the endpoint to the initial point through the recursive relationship. The sum of the optimal cost with the endpoint is the value of $TOTAL_{N-1,N-1}$. By reading the precursor recorded in the MATCH from the endpoint, when we return to the starting point, we can automatically get the optimal path from the endpoint to the initial point: $(i, j) = (i-1, MATCH_{i,j})$, $(i = N-1, \ldots 2, 1, \ j = N-1, \ldots, 2, 1)$. Where i represents the point index on the standard

image of the blood vessel, j represents point index on the constructed image of the blood vessel, and the point pair $P_i, Q_j$ is the optimal matching pair after the solution.

Finally, the optimal matching sequence is obtained.

### 3.5. Surface Reconstruction

The 3D surface can be reconstructed by a series of cross-sectional 3D contours of an ellipse centered on each $(x_i, y_i, z_i)$ point of the blood vessel 3D skeleton. The normal vector $n_i$, of each cross-section plane, $\pi_i$ can be obtained from the tangential vector of the spline curve of the reconstructed 3-D vessel skeleton at the point $(x_i, y_i, z_i)$. Then, the surface normal vector of the point of the contour is perpendicular to $n_i$. For one of the angiographic images, assume that $(u_i, v_i)$ is the ith point of the vessel centerline. $e_i$ and $e\prime_i$ are the two border points of $(u_i, v_i)$, which defines the projected diameter $d_i$. Additionally, $a_i$ represents the normal vector of the projection plane $\pi_{a,i}$ calculated with $e_i, e\prime_i$ and the projective source. Then the vector of the cross line of planes, $\pi_i$ and $\pi_{a,i}$ can be calculated by $l_i = n_i \times a_i$. It is shown in Figure 6 that the length of $l_i$ can be calculated by the following formula:

$$\|l_i\| = d_i \times \left( \sqrt{x_i{}^2 + y_i{}^2 + z_i{}^2} / \sqrt{u_i{}^2 + v_i{}^2 + SID^2} \right), \tag{11}$$

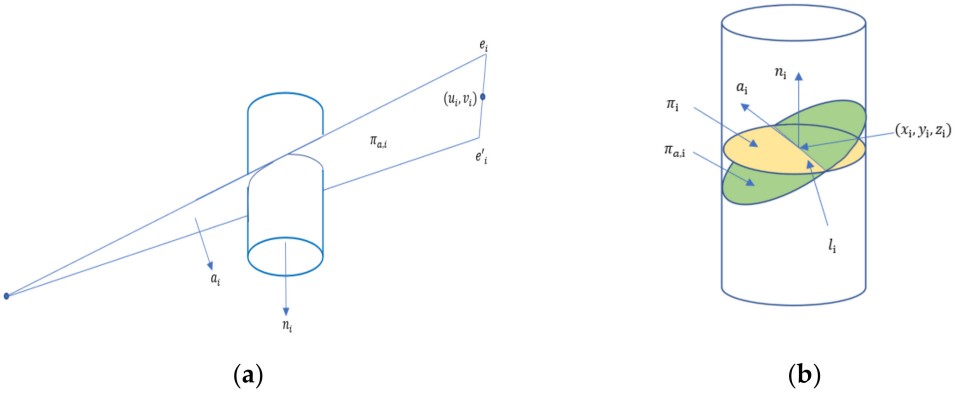

(**a**)    (**b**)

**Figure 6.** (**a**) is the projective plane $\pi_{a,i}$; (**b**) is the intersection line of the ellipse cross-section and projective plane.

Each ellipse is determined by five degrees of freedom. In this paper, the center coordinates are determined, so there are still three degrees of freedom left. We take the longer intersection line as the long axis in the ellipse. So, the direction and length of the long axis have been determined. The complete equation of the ellipse can be obtained by substituting an endpoint of the shorter intersection line into the ellipse.

After obtaining a set of ellipse cross-sections, a series of point clouds can be obtained by sampling the ellipse angle equiangular. Some preprocessing should be done before the point cloud reconstruction: A hemispherical point cloud is added at both ends of the centerline to avoid the truncation of the reconstructed vessels. The sampling points on the intersecting ellipse contour line are fused according to the principle that the distance to the center should remain the same. The point cloud is downsampled according to the density.

Then the point cloud is reconstructed by an implicit surface reconstruction method [22,23].

## 4. Results

For each vascular segment of interest, we choose two images from different angles for reconstruction, each of which is a single static $512 \times 512$ image and its frame rate is 66.67. These images are usually generated by XRA and may pose a risk of radiation exposure [24].

As shown in Table 1, there are three types of clinical data for pole-matching problems in the table. We used these data to reconstruct 3D points. As shown in Figure 7, the solid white line in the blood vessel is the centerline of the blood vessel to be reconstructed.

**Table 1.** Paraments of different datasets.

| Images | Paraments | LAO/RAO (Degree) | CAUD/CRAN (Degree) | SID (mm) | SOD (mm) |
|---|---|---|---|---|---|
| Data.1 | (a) | 45.5 | −28.2 | 1200 | 749 |
| | (b) | −24.8 | −27 | 1147 | 844 |
| Data.2 | (c) | 45.2 | 0 | 1028 | 759.4 |
| | (d) | 0.9 | 33 | 1064 | 766.4 |
| Data.3 | (e) | −21.6 | −21 | 1075 | 782.7 |
| | (f) | 44.3 | −28.8 | 1118 | 785 |

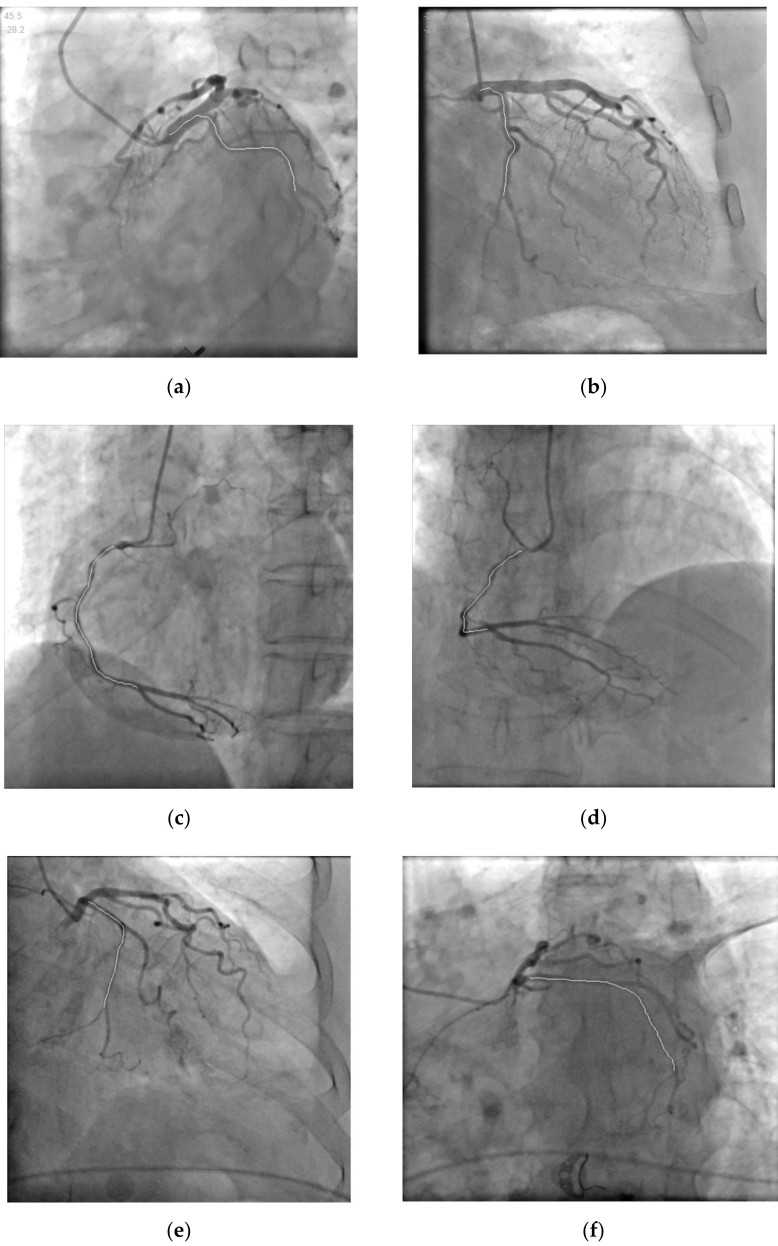

**Figure 7.** Clinical contrast images for reconstruction. (**a**,**b**) are Data.1; (**c**,**d**) are Data.2; (**e**,**f**) are Data.3.

To evaluate the above systematic parameters optimization algorithm, Optimize 1 is a heuristic simulated annealing algorithm, Optimize 2 is a traditional LM algorithm, and Optimize 3 is a parameter self-adjusting LM algorithm used in this paper.

Table 2 evaluates the systematic parameters optimization algorithm by some kind of index. Figure 8. shows the accuracy of the epipolar line of calculating by the optimized systematic parameters. The data used in Figure 8 is Data.1 because its initial error is the biggest.

**Table 2.** Objective function optimization index statistics.

| Methods | Items | Before Optimization (mm) | After Optimization (mm) | Time (ms) | Number of Iterations |
|---|---|---|---|---|---|
| Data.1 | Optimize 1 |  | 0.00188 | 16303 | 695 |
|  | Optimize 2 | 77.5235 | 1.69251 | 690 | 164 |
|  | Optimize 3 |  | $9.99 \times 10^{-11}$ | 238 | 54 |
| Data.2 | Optimize 1 |  | $4.51 \times 10^{-6}$ | 15929 | 695 |
|  | Optimize 2 | 1853.77 | $1.83 \times 10^{-5}$ | 589 | 157 |
|  | Optimize 3 |  | $8.94 \times 10^{-11}$ | 119 | 27 |
| Data.3 | Optimize 1 |  | $1.86 \times 10^{-7}$ | 16027 | 695 |
|  | Optimize 2 | 44539.9 | 1.65073 | 522 | 130 |
|  | Optimize 3 |  | $3.82 \times 10^{-11}$ | 142 | 32 |

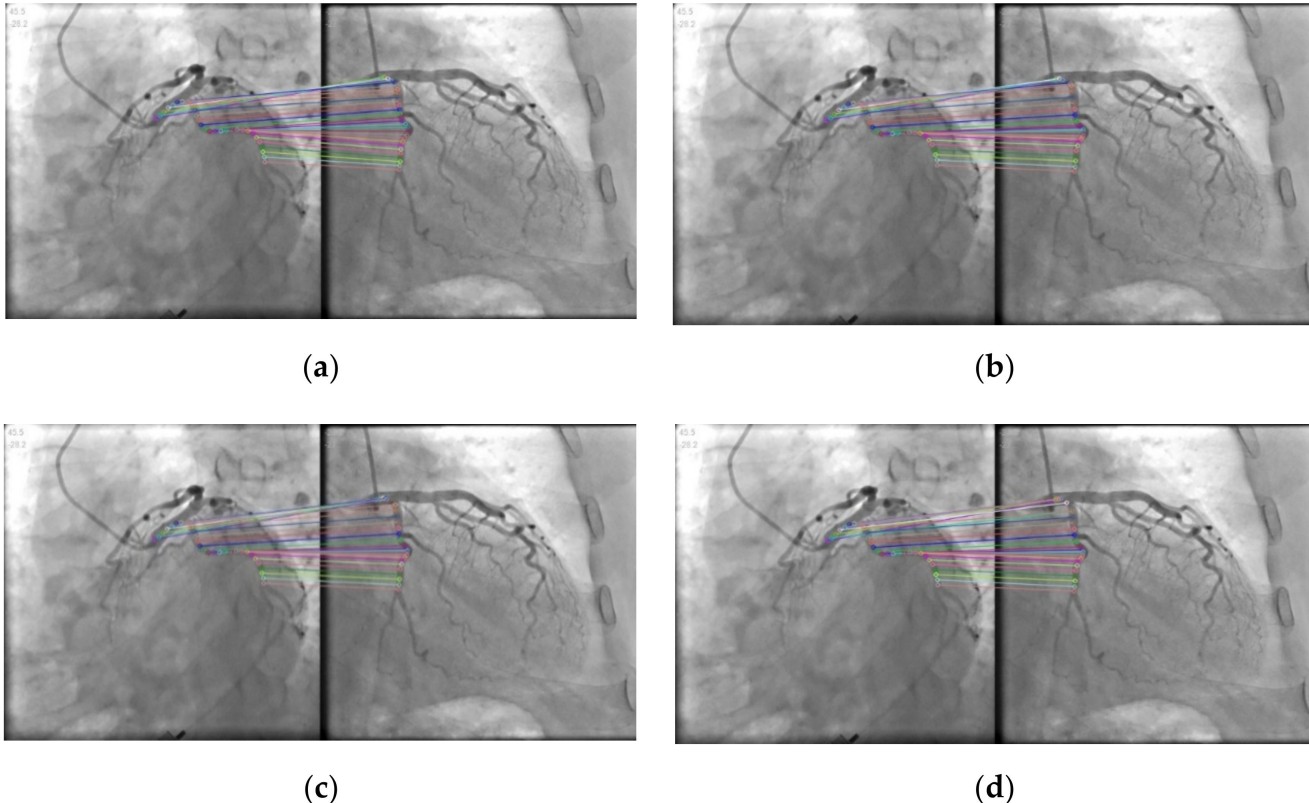

**Figure 8.** Matching results after different optimization. (**a**) is the result of Match 1; (**b**) is the result of Match 1; (**c**) is the result of Match 1; and (**d**) is the result of Match 1.

It can be seen that the accuracies of the epipolar line after all the three kinds of optimization are similar for matching. So, the main concerns are optimization efficiency and robustness.

In Table 2, because of the cooling strategy of Optimize 1, the fixed 695 times iterations, which will waste a lot of time, and the instability of the results limits its clinical application.

As for Optimize 2, it cannot achieve the target error in Data.1 and Data.2, which means it will fall into the local optimal solution. The convergence of Optimize 3 is far superior to other methods, reaches the ideal limit error, and can meet the high-precision requirements in clinical practice. At the same time, the iterations number and optimization time of Optimize 3 is also the smallest, which has greater advantages compared to other methods.

In order to evaluate the effect of our proposed algorithm on improving the accuracy of the 3D reconstruction, the directly matching method relying on the order of points on blood vessels is recorded as Match 1, the dynamic programming algorithm with fixed step size is recorded as Match 2, and the dynamic programming algorithm with self-adaption step size according to CBOCD proposed in this paper is recorded as Match 3. Finally, we assume that the traditional method only uses epipolar line constraint as Match 4.

Clinical studies indicate that to evaluate the accuracy of matching point pairs, it is necessary to compare the same feature points matching in the blood vessel segment. As shown in Figure 8, from Match 1 to Match 4, the smoothness of conjugate point pairs decreased gradually. The result of Match 4 even has some mismatches. In other words, when there are multiple intersections between the vessel and the epipolar line, the matching points that do not conform to the smoothness of blood vessels are selected by mistake. So, Match 4 cannot be used in the 3D reconstruction of blood vessels.

Table 3 shows the error statistics of the Euclidean distance between the points on the original contrast image and the back-projection image and the distance between the reconstructed 3D point and the corresponding two epipolar lines. It can be seen that Match 2 has greatly improved the accuracy of 3D reconstruction from each index. This precision can meet the requirements of medical applications. As for Match 3, it sometimes reduces the reconstruction error to fifty percent of the error with Match 2, like Data.1 and Data.2. However, in Data.3, the reconstruction error between Match 2 and Match 3 is almost not changed. The reason is that constructed CBODE of the blood vessels is almost two. So the step size of dynamic programming in Match 3 can be regarded as the fixed step. The proposed method is evidently effective and robust for building matching relationships for vascular branches in coronary angiograms.

**Table 3.** Statistics with the error of 3D reconstruction.

| Methods | Items | | Maximum | Minimum | Mean | RMS | 3D |
|---|---|---|---|---|---|---|---|
| Data.1 | (a) | Match 1 | 1.90582 | 0.00175 | 0.79173 | 0.49422 | 83.1266 |
| | | Match 2 | 0.70741 | 0.00123 | 0.13104 | 0.1651 | 4.2336 |
| | | Match 3 | 0.6653 | $2.02 \times 10^{-4}$ | 0.06917 | 0.12584 | 1.99264 |
| | (b) | Match 1 | 1.74575 | 0.0016 | 0.73019 | 0.4522 | / |
| | | Match 2 | 0.66112 | 0.00116 | 0.12136 | 0.1535 | / |
| | | Match 3 | 0.62295 | $1.89 \times 10^{-4}$ | 0.06439 | 0.11788 | / |
| Data.2 | (c) | Match 1 | 5.66455 | 0.0334 | 2.72141 | 1.83636 | 824.806 |
| | | Match 2 | 1.06809 | 0.00136 | 0.22034 | 0.29041 | 10.4045 |
| | | Match 3 | 0.80592 | $4.21 \times 10^{-4}$ | 0.09791 | 0.19193 | 3.64186 |
| | (d) | Match 1 | 5.76568 | 0.0333 | 2.76739 | 1.87068 | / |
| | | Match 2 | 1.07169 | 0.00135 | 0.22205 | 0.29158 | / |
| | | Match 3 | 0.81107 | $4.1 \times 10^{-5}$ | 0.09869 | 0.19298 | / |
| Data.3 | (e) | Match 1 | 3.31963 | 0.00151 | 1.35754 | 1.09306 | 350.884 |
| | | Match 2 | 0.09431 | $5.77 \times 10^{-5}$ | 0.02394 | 0.01995 | 0.11051 |
| | | Match 3 | 0.079089 | $1.77 \times 10^{-4}$ | 0.02334 | 0.0186 | 0.10128 |
| | (f) | Match 1 | 3.51223 | 0.00161 | 1.43727 | 1.15846 | / |
| | | Match 2 | 0.09964 | $6.03 \times 10^{-5}$ | 0.02519 | 0.02092 | / |
| | | Match 3 | 0.08201 | $1.86 \times 10^{-4}$ | 0.02456 | 0.01947 | / |

Figure 9 is the process of constructing the CBOCD. Where there is overlap or parallel to the contrast line, it is correctly marked as shown in Figure 9a. The convex arcs on the

blood vessels are also detected correctly by using the method proposed in this paper, as shown in Figure 9b. The low brightness segments were also correctly marked as shown in Figure 9c. However, it still has some false recognition due to the influence of background brightness, as mentioned above.

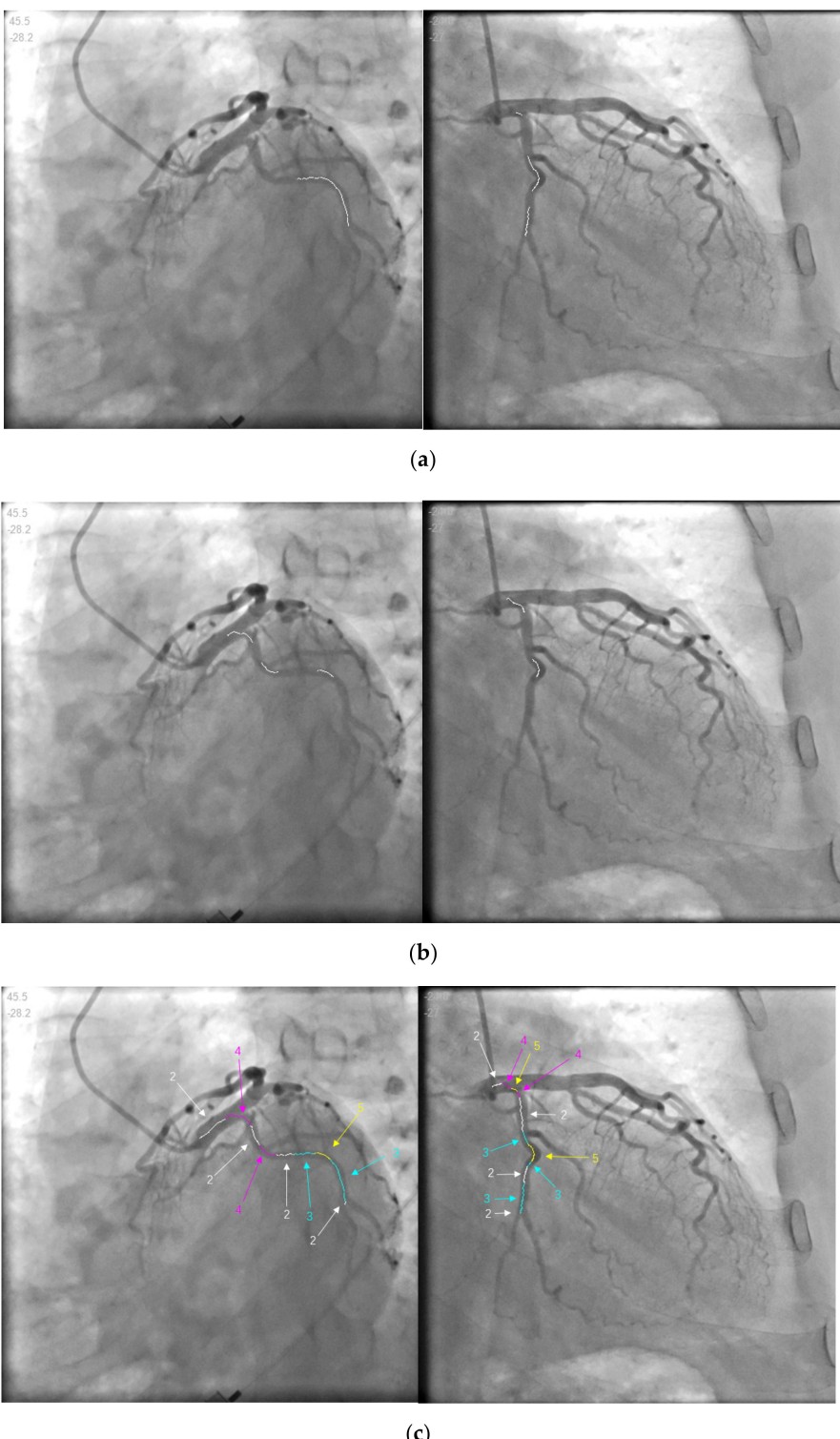

**Figure 9.** The CBODE of Data 1. (**a**) is the Brightness OPE; (**b**) is the Curvature OPE; (**c**) is the content of CBODE.

Multi-view geometry theory shows that the constructed matching relationship can be used for the 3D reconstruction of blood vessels. The correctness of the matching relationship can be effectively visualized from the results. Figure 10 highlights the result of the 3D reconstruction results of Data.1. The reconstructed centerline of blood vessels smoothed by non-uniform rational basis splines (NURBS) is shown in Figure 10a. Figure 10b is the reconstructed surface after head and tail filling. Because the radio of the head and tail of the blood vessel suddenly drop to zero, we draw a hemisphere with the radius of the nearest point and add it to the head and tail of the vessel.

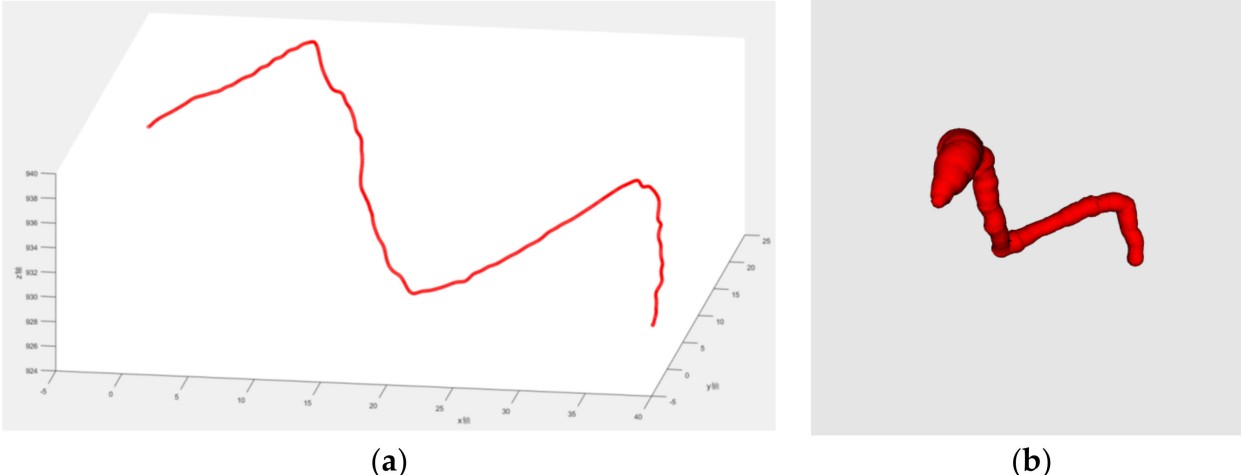

(**a**)  (**b**)

**Figure 10.** The result of the 3D reconstruction results is based on the proposed method. (**a**) is the reconstructed centerline; (**b**) is the reconstructed surface after head and tail filling.

## 5. Discussion and Conclusions

In this paper, we proposed a new approach for the 3D reconstruction of the coronary artery, which is based on a curve descriptor and projection geometry-constrained vasculature matching and the parameter-adjusting LM optimization method for system parameters. Experiments show that our method can effectively improve the accuracy of blood vessel 3D reconstruction and solve some of the potential problems raised in this paper. The curve descriptor used with dynamic programming improves the smoothness while ensuring accuracy. The parameter-adjusting LM optimization has stability and global convergence while being solved efficiently.

We construct curve descriptors by brightness and convex arc. However, the brightness of the measurement method proposed in this paper cannot completely remove the influence of background noise. Therefore, how to accurately reduce the vascular brightness determined by the contrast medium content is one of the focuses of future research. At the same time, the detection method of convex arc in the curve is also developing rapidly. In the future, we will try to Choose a more appropriate and more accurate convex arc detection method.

**Author Contributions:** Conceptualization, J.T. and P.Q.; methodology, S.X.; validation, J.T.; formal analysis, F.W.; writing—original draft preparation, S.X.; writing—review and editing, P.Q.; visualization, F.W.; funding acquisition, J.T. All authors have read and agreed to the published version of the manuscript.

**Funding:** This research was funded by National Natural Science Foundation of China grant number 11304382; Science and Technology program of Jinhua Science and Technology Bureau grant number 2020-3-001, 2020-3-007; Science and Technology program of Yiwu Science and Technology Bureau grant number 20-3-239.

**Conflicts of Interest:** The authors declare no conflict of interest.

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
