# Peer review of "3D Reconstruction with Coronary Artery Based on Curve Descriptor and Projection Geometry-Constrained Vasculature Matching"

_information, doi:10.3390/info13010038_

Round 1

Reviewer 1 Report

The authors proposed 3D reconstruction of coronary artery from coronary angiography based on curve descriptor and projection geometry constrained vasculature matching. There are several important matters which require revisions.

  1. Explanation of the abbreviations first used in the paper; for example, LM, XRA...
  2. Clinical importance and significance are unclear. Experienced cardiologists decide the direction of the projection because they have already understood 3D structure of coronary artery. Do the authors aim quantitative analysis of the degree of stenosis?
  3. Description of the details of the angiography data is required. Frame rate, number of the pixels of a single image, and so on.
  4. English should be edited.

Author Response

Response to Reviewer 1 Comments

Point 1: Explanation of the abbreviations first used in the paper; for example, LM, XRA...

Response 1: We have carefully checked our paper and added explanations of all abbreviations. And LM algorithm means Leveberg-Marquardt algorithm; XRA means X-ray angiography…

Point 2: Clinical importance and significance are unclear. Experienced cardiologists decide the direction of the projection because they have already understood the 3D structure of coronary artery. Do the authors aim quantitative analysis of the degree of stenosis?

Response 2:

  1. a) Realize the three-dimensional visualization of the blood vessel segment of interest in the coronary artery tree. The doctor can better check the shape and size of the blood vessel through interactive operations such as rotation and zoom, and enhance the understanding of the overall situation of the blood vessel, thereby reducing the need for diagnostic examination The number of radiography reduces the cost of medical treatment and at the same time alleviates the patient’s pain;
  2. b) Realize the three-dimensional quantitative analysis of coronary arteries [3], and accurately and quantitatively describe the length, diameter, direction and blood flow volume of the coronary arteries and other important parameters, which will help describe the condition more accurately and improve the accuracy of clinical diagnosis sex. Especially for the location of vascular stenosis and the judgment of the degree of stenosis, it is difficult to obtain accurate values ​​of these parameters with traditional methods in the past;
  3. c) Through the three-dimensional virtual display of the blood vessel segment of interest in the coronary tree, the best observation angle for the location of the vascular lesion can be calculated, so that new angiographic images can be obtained based on the best observation angle, which can better Examine this section of the blood vessel to assist the doctor in further accurate diagnosis and treatment;
  4. d) Combined with other cardiovascular images, perform linkage analysis with the reconstructed 3D model. For example, by fusing the reconstructed 3D model of the coronary artery with the IVUS image, while understanding the external 3D information of the suspected vascular disease location, you can also observe the details of the inside of the blood vessel at that location to obtain more blood vessel information, to better understand the patient is diagnosed.

Point 3: Description of the details of the angiography data is required. Frame rate, number of the pixels of a single image, and so on.

Response 3:

Frame Time: 66.666667

Image Pixel Spacing :0.308 mm/pixel

Rows:        512

Columns:   512

This information is listed at the beginning of the experimental results in the revised version of our paper.

Point 4: English should be edited.

Response 4: We have asked dr. Tong, who’s a well-established expert, to polish our paper. Please see if the revised version met the English presentation standard.

Reviewer 2 Report

Overall well written manuscript illustrating a novel method based on curve descriptor and projection geometry constrained vessel matching. The research methods are rigorous and clear, and the study findings are presented in a clear and exhaustive manner.

My main suggestion is to discuss in more detail how and to what extent the system described in the study, or any derivatives thereof, could be successfully and conveniently integrated into clinical practice (e.g., into existing imaging systems for interventional cardiology applications) to the advantage of both patients and healthcare operators.

Furthermore, it would be important to briefly discuss how the system presented in the study could work under relatively high image noise conditions - this is relevant e.g., for operation with very large patients and/or to reduce ionizing radiation exposure to patients and staff during interventional procedures. A useful reference (which should be added as a reference) to put the issue of radiation protection into context is the following: Bastiani L et al, JAMA Netw Open. 2021 Oct 1;4(10):e2128561. doi: 10.1001/jamanetworkopen.2021.28561 

Author Response

Response to Reviewer 2 Comments

Point 1: My main suggestion is to discuss in more detail how and to what extent the system described in the study, or any derivatives thereof, could be successfully and conveniently integrated into clinical practice (e.g., into existing imaging systems for interventional cardiology applications) to the advantage of both patients and healthcare operators.

Response 1: To more easily integrate our algorithms into clinical practice, our team has built a complete set of cardiovascular disease management systems, including nursing, surgery, coronary imaging, and postoperative follow-up. The system is connected to the HIS and the PACS system which provides great convenience for the entire clinical practice. The main purpose of the method proposed in the article is the linkage analysis with the other cardiovascular images. For example, by fusing the reconstructed 3D model of the coronary artery with the IVUS image, while understanding the external 3D information of the suspected vascular disease location, you can also observe the details of the inside of the blood vessel at that location to obtain more blood vessel information, to better understand the patient is diagnosed.

 Point 2: Furthermore, it would be important to briefly discuss how the system presented in the study could work under relatively high image noise conditions - this is relevant e.g., for operation with very large patients and/or to reduce ionizing radiation exposure to patients and staff during interventional procedures. A useful reference (which should be added as a reference) to put the issue of radiation protection into context is the following: Bastiani L et al, JAMA Netw Open. 2021 Oct 1;4(10):e2128561. doi: 10.1001/jamanetworkopen.2021.28561 

Response 2: We carefully read the paper you mentioned and added it as a reference to our paper. Large background noise has a great influence on the reconstruction result. At present, we perform reconstruction by removing these noisy images through pre-processing. In the discussion in the last chapter of our paper, we mentioned that the future work is mainly to eliminate the influence of background noise.

Round 2

Reviewer 1 Report

The authors satisfactory revised the manuscript according to the previous comments. Now the manuscript should be accepted.